# The Chemistry, Recrystallization and Thermal Expansion of Brannerite from Akchatau, Kazakhstan

**DOI:** 10.3390/ma16041719

**Published:** 2023-02-18

**Authors:** Ruiqi Chen, Oleg I. Siidra, Vera A. Firsova, Angel Arevalo-Lopez, Marie Colmont, Valery L. Ugolkov, Vladimir N. Bocharov

**Affiliations:** 1Department of Crystallography, Institute of Earth Sciences, St. Petersburg State University, 199034 St. Petersburg, Russia; 2Université Lille, CNRS, Centrale Lille, Université Artois, UMR 8181, UCCS, Unité de Catalyse et Chimie du Solide, F-59000 Lille, France; 3Institute of Silicate Chemistry, Russian Academy of Sciences, 199034 St. Petersburg, Russia; 4Geomodel Resource Center, St. Petersburg State University, 199034 St. Petersburg, Russia

**Keywords:** brannerite, metamict minerals, recrystallization of metamict minerals, thermal analysis, thermal expansion, radioactive minerals, matrix for HLW immobilization

## Abstract

Numerous studies expose the potential of brannerite to become a good matrix, concentrating fission products and actinides. Minerals can complement the data collected from the synthetic materials and offer an advantage of a long-time exposure to radiation. Natural metamict brannerite from Akchatau, Kazakhstan, and its annealed sample were studied by EPMA, Raman spectroscopy, TGA, DSC, XRD and HTXRD. The radioactivity of pristine and annealed samples of brannerite was measured. Brannerite from Akchatau is characterized by the absence of significant amounts of REE and yttrium. The studied brannerite regains its structure at a temperature ~650 °C, revealed by the HTXRD and DSC. HTXRD was also performed on the annealed recrystallized brannerite. The thermal expansion for brannerite has been determined for the first time. The brannerite structure expands anisotropically with temperature increase. All the thermal expansion coefficients are positive except for *α_β_*. The decreasing beta parameter indicates a “shear structural deformation“. The angle between the 1st axis of the tensor and the crystallographic *a* axis decreases with the increase of the temperature. The structure expands mostly in the *α*_11_ direction, approaching the bisector of the *β* angle. Brannerite has a low CTE at room temperature—*α_v_* = 16 × 10^−6^ °C^−1^, which increases up to 39.4 × 10^−6^ °C^−1^ at 1100 °C. In general, the thermal stability of brannerite is comparable to that of the other perspective oxide radioactive waste-immobilizing matrices (e.g., *Ln*_2_Zr_2_O_7_, CePO_4_, CaTiO_3_, CaZrTi_2_O_7_). The calculated thermal expansion of brannerite and the understanding of its underlying crystal chemical mechanisms may contribute to the behavior prediction of the material (both metamict and crystalline) at high temperatures.

## 1. Introduction

Brannerite is a uranium titanate mineral with an ideal formula of UTi_2_O_6_. It ideally contains 62.8 wt.% UO_2_ [1,2]. The high content of uranium makes brannerite an exploitable uranium resource. It contains uranium in a multivalent state due to its origin [3] and redox reactions are caused by the radioactive decay. Therefore, tetra-, penta- and hexavalent uranium are able to incorporate into the natural and synthetic brannerite [4,5]. ThO_2_ is a common component of the natural brannerite, along with up to 8 wt.% of rare earth elements and a small amount of CaO, Fe_2_O_3_, PbO and Al_2_O_3_ [3,6]. Due to the hydration and alteration, the significant amounts of Si and other elements are able to be incorporated into the structure [7]. Thus, the mineral has the general formula of *AB*_2_O_6_, where the *A* site can be occupied by U, Ca, Th, Y and REE, and the *B* site is mainly occupied by Ti, Nb, Si, Fe and Al [1,3,7]. 

Deciphering the crystal structure of brannerite has encountered difficulties. Natural brannerite specimens are often metamict because of the radiation damage. However, Szymanski and Scott [2] successfully determined the crystal structure of the synthetic UTi_2_O_6_, indicating the monoclinic symmetry and space group *C*2/*m*. Both the U and Ti atoms are in distorted octahedral coordination. TiO_6_ octahedra are linked one with each other via a common edge, thus forming a TiO_2_ layer parallel to the *ab* plane. Layers are interconnected via UO_6_ octahedra, as shown in Figure 1a.

The highly radioactive, uranium-bearing mineral was first found in a gold placer in Idaho, USA [8]. As an accessory mineral, brannerite is very stable and can be found in various geological environments, including granites, granitic pegmatites and hydrothermally altered sedimentary rocks. The deposit Crocker’s Well in South Australia is related to the late tectonic granitoid intrusions, pegmatitic brannerite is usually found in association with Ti oxides, mainly rutile and U–Th-rich silicates [3]. Low-temperature Au deposits at Witwatersrand (South Africa) may also be a source of brannerite, where brannerite is considered as a secondary uranium mineral, it appears as an inclusion in detrital minerals, linings around pyrobitumen nodules and micro veins in gold-bearing horizons [9]. The occurrence of brannerite in the Central African Orogenic Belt is related to post-magmatic metamorphism, and the minerals associated with brannerite are monazite, uraninite, zircon, albite, calcite, chlorite, apatite, U-silicates and iron oxides [10]. The Hüttenberg deposit is an Eastern Alps Cretaceous siderite deposit where brannerite exhibits euhedral prismatic crystals and shows the evidence of hydration, metamictization and alteration to anatase [11]. Brannerite is reported also in altered ores at the uranium deposit Komsomolskoye, Central Kazakhstan [12]. In this work, reported below, the brannerite sample (Figure 1b) originates from the Akchatau, Kazakhstan, where the W-Mo deposits are hosted within leucocratic granites of the Permian age [13].

Brannerite is a refractory titanate, and extraction of the uranium requires more intense leaching conditions than typical uranium oxide ores. However, on the other side, the stable titanate structure can be used in material science. Brannerite-based synthetics have been widely studied. Much research exposes brannerite’s potential to become a matrix, concentrating fission products and actinides. It, therefore, can be used for radioactive waste immobilization. Exploration of brannerite synthesis techniques can provide new insights for the immobilization of radionuclides [14]. The solid phase method, commonly used for ceramic syntheses, is based on the firing of a mixture of uranium and titanium dioxides at temperatures in the range 1300–1400 °C [15]. The crystals used by Szymanski and Scott [2] for the structure determination were grown by a cryolite fusion technique. The wet chemistry route is also adopted for brannerite synthesis [14,16].

To avoid the leakage of hazardous elements from the complex ceramics to the environments, it is important to determine the thermal expansion characteristics of the matrix. Minerals can complement the data collected from the synthetic materials and offer an advantage of a long-term exposure to the radiation. The aim of this work was to study a natural brannerite; however, since the mineral samples are metamict, their recrystallization was studied. The thermal expansion of the crystalline brannerite is reported for the first time.

## 2. Materials and Methods

Unheated (metamict) and heated sample portions were investigated by electron probe microanalysis (EPMA), Raman spectroscopy (RS), X-ray powder diffraction (XRD and high-temperature XRD), thermal analysis (TGA-DSC) and gamma-ray spectrometry.

### 2.1. Sample

Samples of brannerite were obtained from the Mineralogical Museum of Saint Petersburg University, Russia. The mineral sample is from the Akchatau (Kazakhstan) greisen W-Mo deposit. Brannerite is represented by metamict grains up to 2 mm in length (Figure 1b). Brannerite grains were handpicked and investigated under the optical microscope to avoid the presence of the impurities of other minerals.

The heating of metamict brannerite grains for the subsequent studies was carried out in platinum crucibles at a temperature of 1200 °C in a laboratory furnace produced by Nabertherm.

### 2.2. Quantitative EMPA

The metamict and heated brannerite grains were prepared by mounting them in an epoxy resin. The samples were studied using the scanning electron microscope Hitachi S-3400N, equipped with a spectrometer, Oxford Instruments X-Max 20, for energy-dispersion analysis. Photos of the samples were taken in the backscattered electron mode (BSE). Quantitative analyses were performed with an accelerating voltage of 20 kV and a beam current of 1.8 nA. Acquisition time is 30 s, with a resolution up to 4 nm. The following standards were used for the quantification: MAC (Micro Analysis Consultants Ltd., United Kingdom); Geller reference standards (Geller microanalytical laboratory)

### 2.3. Raman Spectroscopy

The metamict and annealed grains of brannerite were placed on the stage of an Olympus BX41 microscope, equipped with a Horiba Jobin-Yvon LabRam HR800 Raman spectrometer. Raman spectra were excited by an Ar^+^ laser (457~514 nm) in the range 100–4000 cm^−1^. Baseline correction was used to improve the signal-to-noise ratio.

### 2.4. X-ray Powder Diffraction

The heated and metamict samples were ground for the powder X-ray diffraction studies. Preliminary X-ray diffraction (XRD) analysis indicated that the investigated samples are metamict. The sample calcined in air at a temperature of 1200 °C for 6 h produced well-crystallized phases. Preliminary XRD measurements at room temperature were carried out using a Rigaku Miniflex II diffractometer (Co-Kα, 30 kV, 15 mA).

The high-temperature X-ray (HT-XRD) experiments were performed using a Rigaku Ultima IV diffractometer with a thermal attachment (Co-Kα, 40 kV and 30 mA, reflection geometry, D/teX ultra-high-speed detector, air atmosphere, 2θ = 10–80°, temperature range 25–1200 °C, step size 100 °C for the temperature range 20–600 °C and 25 °C for 600–1200 °C). HT-XRD measurements performed for metamict samples demonstrate the evolution of the structure during recrystallization, whereas the results obtained from the annealed sample provided the data for the thermal expansion calculation.

### 2.5. Thermal Investigations

Thermal analysis of the metamict brannerite was performed in air with a 20 °C/min heating rate, the powder sample was heated up to 1300 °C and subsequently cooled to 300 °C at the same rate. TGA and DSC curves were recorded using TA Instruments STA 429 CD.

### 2.6. Radioactivity Measurements

Measurements of the radioactivity exhibited by the studied brannerite were carried out. Gamma-ray spectra for 3.6 g of a sample were obtained and processed using the gamma-spectrometer MKGB-1. The activity and specific activity of gamma-emitting radionuclides ^232^Th, ^226^Ra, ^137^Cs и ^40^K in the sample were determined. The radioactivity of the powder sample was repeatedly examined after its calcination in air at a temperature of 1200 °C for 6 h.

## 3. Results

### 3.1. Chemical Composition and Phase Purity

Two pristine grains (***Brn1*** and ***Brn2***) and one calcined grain (***Brn3***) were investigated by EPMA, the BSE image is shown in Figure 2.

The unheated samples exhibit the following compositional ranges: 51.9–58.3 wt.% UO_2_, 32.8–36.6 wt.% TiO_2_, 3.9–6.6 wt.% ThO_2_. Additional constituents include, on average, 0.8 wt.% Fe_2_O_3_, 0.8 wt.% CaO, 0.9 wt.% PbO, 0.8 wt.% SiO_2_, 0.5 wt.% Na_2_O. The chemical analyses (average of 5 analyses for ***Brn1***, 15 analyses for ***Brn2*** and 7 analyses for ***Brn3***), the ranges and the numbers of *a.p.f.u.* are given in Table 1. The standards used are Ti-metal, Fe-metal, Nb-metal, SiO_2_, U_3_O_8_, ThO_2_, CaSO_4_, NaCl and PbS.

According to the analysis, the Akchatau brannerite is close to the stoichiometric brannerite and contains a significant amount of thorium to form a solid solution with thorutite, ThTi_2_O_6_. Surprisingly, rare earth elements are not detected in studied samples. Total measured contents are close to 100%, water was not detected. The presence of small amounts of lead may originate as a decay product of the uranium series.

***Brn1*** is a heterogenous brannerite, Figure 2 shows the contrast in the BSE intensity, indicating that the elements are unevenly distributed. Some areas show evidence of the alteration. The elongated section, spreading along the fracture, appears as “darker” BSE intensity in comparison with the overall grain. The analytic total of this area is reduced to ~95%. The low amount shows that some lightweight, undetectable components concentrate here; meanwhile, the weight percentage of SiO_2_ increases up to 0.5%. Other areas in ***Brn1*** do not contain SiO_2_. Several rhombic crystals (5–8 µm) were identified as uranium-rich pyrochlore with ~10% H_2_O (Figure 2a).

In addition to inclusions, little “bright white” spots of 0.4–3 µm in size can be seen in both grains (Figure 2, ***Brn1*** and ***Brn2***). Their composition cannot be determined unambiguously due to the capture of other elements from the surrounding brannerite matrix. Table 2 shows their average chemical composition with totals normalized to 100. High contents of PbO and S suggest it to be galena, PbS.

The grain ***Brn2*** is even less homogeneous compared to ***Brn1***, Table 1 demonstrates its chemical composition based on 15 spot analyses. The grain ***Brn2*** is characterized by the linearly altered zones, where the content of SiO_2_ reaches 1.5–1.7 wt.% (Figure 2, ***Brn2***).

The formula for ***Brn1*** was initially calculated based on O = 6: (U_0.90_Th_0.09_Ca_0.05_Pb_0.02_)_∑=1.05_(Ti_1.92_Fe_0.02_Si_0.02_)_∑=1.96_O_6_□^+^_0.12_. Considering the presence of the alteration zones with hydroxyl, the calculation accuracy based on 6 O *apfu* is rather poor. Thus, the formulas were calculated based on *B = 2 apfu*: (U_0.91_Th_0.09_Ca_0.05_Pb_0.02_)_∑=1.06_ (Ti_1.94_Fe_0.04_Si_0.02_)_∑=2_O_6.06_.

Formulas for ***Brn2*** are: (U_0.87_Th_0.09_Na_0.03_Ca_0.06_Pb_0.01_)_∑=1.06_(Ti_1.86_Fe_0.02_Si_0.10_)_∑=1.06_O_6_□^+^_0.28_ and (U_0.87_Th_0.08_Na_0.07_Ca_0.06_Pb_0.01_)_∑=1.09_(Ti_1.86_Fe_0.04_Si_0.10_)_∑=2_O_5.99_, calculated on the same basis described above, respectively.

It can be seen that the chemical composition of brannerite fulfills the stoichiometry *AB*_2_O_6_, where *A* = U and Th; *B* = Ti and Fe. The localization of Si is not clear yet. It is presumed to occupy part of the *B* site as its atomic radius is closer to that of Ti rather than U.

Another grain, ***Brn3***, was calcined in the atmosphere at 1200 °C for 4 h. Its chemical composition is also listed in Table 2. The chemical composition of the annealed brannerite is similar to that of the metamict grain. Calcined brannerite apparently becomes homogeneous, although the texture shows increased porosity typical for recrystallized metamict minerals. Numerous bright spots (0.2–1 µm) are observed in the enlarged BSE image. However, their chemical content has not been successfully determined due to their small size.

### 3.2. Raman Spectroscopy

Raman spectroscopy was used to investigate both the metamict and annealed brannerite. Frost and Beddy [17] compared the Raman spectra of brannerite and uranyl oxyhydroxide hydrates; they attributed the observed bands to the (UO_2_)^2+^ and TiO stretching and bending vibrations, as well as U–OH and O-Ti-O bending vibrations. Charalambous and coauthors [7] examined partially metamict brannerite, the Raman spectra of metamict minerals exhibited broad bands in the range of 50–1100 cm^−1^. Zhang and coauthors [18] obtained Raman spectra of natural and synthetic brannerite and recognized some incorrect Raman assignments performed in previous works.

We obtained Raman spectra in the 90–3500 cm^−1^ range for both metamict and annealed samples. The Raman spectrum of the metamict brannerite shows only a few broad and overlapping bands (Figure 3a). Raman bands are observed only in the range 90–1200 cm^−1^.

The mineral structure is heavily damaged. However, the Raman spectra of most areas perform neither O-H stretching of water molecules and hydroxyl groups at approximately 3500 cm^−1^ nor the O-H-O bending bands at around 1600 cm^−1^. O-H bands are observed only in the areas close to the cracks (Figure 2c), which indicates the partial hydration and alteration.

Annealed brannerite grains are more homogeneous. The vibrational modes are very intense and sharp in the Raman spectra (Figure 3b). The brannerite spectrum is in excellent agreement with the Raman spectrum of the annealed brannerite provided in the previous work of Zhang et al. [18]. The comparison of the obtained data with the reference data is provided in Table 3.

### 3.3. X-ray Powder Diffraction

To identify the recrystallization temperature and to observe, in situ, the crystallization of brannerite, X-ray powder diffraction was performed for the metamict sample. Figure 4 demonstrates the structural evolution of the metamict brannerite with increasing temperature. Pt holder diffraction peaks and unknown impurity were observed during the entire heating process.

Previous research shows that the temperature of the brannerite from Crocker Well (South Australia) recrystallization is in the range of 800–900 °C [7]. Adler and Puig [20] determined the start of brannerite recrystallization above 550 °C. The brannerite from El Cabril mine (Spain) completely recrystallizes between 900 °C and 1100 °C, but the crystallization process starts earlier, in the range 500–700 °C [21]. However, our in situ studies reveal that brannerite from Akchatau recrystallizes at a temperature as low as 625–650 °C. At temperatures above 625 °C, several diffraction peaks quickly evolve, indicating the start of the recrystallization process.

There are some previously published data showing that TiO_2_ and U_3_O_8_ can be produced during the recrystallization of the brannerite-type material, when the stoichiometric material is heated in the air [15]. Uranium oxides appear at ~800 °C: diffraction lines 25°, 31.5° and 40° 2θ can be assigned to U_3_O_8_ (ICSD-51525). This phase, however, disappears at ~900 °C and evolves into the uraninite-type structure UO_2_ (ICSD-61576). It is known that U_3_O_8_ decomposes into UO_2_, with the temperature increase even under mildly reducing conditions [22,23].

Figure 5 shows the ratio of thus formed U_3_O_8_ and UO_2_ with the temperature increase. The diffraction pattern of the sample after cooling shows that the oxides may present only at the levels below the detection limits. In addition, the rutile phase is also noted in the annealed sample. Rutile, TiO_2_, is typically associated with brannerite-rich ores in natural environments. The R-factors calculated by the Rietveld refinement increase with the temperature rise and fluctuate in the range of 6.2–12.6%.

### 3.4. Thermal Expansion

High-temperature powder diffraction experiments were performed on the annealed brannerite (Figure 6). The program “Rietveld to tensor” [24] was used to analyze the XRD data and to calculate the thermal expansion.

Lattice parameters of brannerite were calculated using the Rietveld Method. The structure expands mostly along the *a* axis, while along the *b* axis the changes are very minor. Unit cell parameters *a*, *b* and *c* increase in the range 9.82–9.92 Å, 3.74–3.77 Å and 6.88 to 6.94 Å, respectively. The *β* angle decreases slightly between 118.6 and 118.1°. The unit cell volume increases from 221.89 to 228.83 Å^3^ (Figure 7). The temperature dependence of the unit cell parameters was approximated by polynomials of the second order:*a*(*T*) = 9.8212(14) + 0.0348 (68)·10^–3^·*T* + 0.0489(67) ·10^–6^·*T*^2^*b*(*T*) = 3.7407(28) + 0.0166 (25)·10^–3^·*T* + 0.0027(13) ·10^–6^·*T*^2^*c*(*T*) = 6.8839(13) + 0.0454(13)·10^–3^·*T* + 0.0056(61) ·10^–6^·*T*^2^*β*(*T*) = 118.634(22) − 0.06(10)·10^–3^·*T* − 0.489(67) ·10^–6^·*T*^2^*V*(*T*) = 221.005(62) + 3.2144(13)·10^–3^·*T* + 2.6698(43) ·10^–6^·*T*^2^,(1)

*T*-temperature.

**Figure 7 materials-16-01719-f007:**
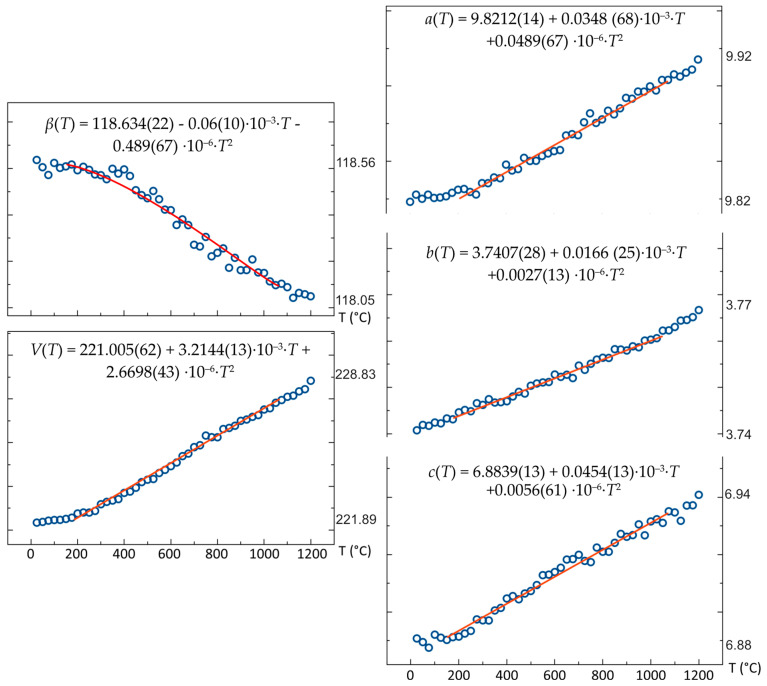
Unit cell parameters of crystallized brannerite in relation to temperature.

Table 4 shows the obtained values. The value of the volumetric expansion α_v_ is the sum of the eigenvalues *α*_11_, *α*_22_ and *α*_33_. The linear expansion along the crystallographic axis *b* is *α*_22_. All the thermal expansion coefficients are positive except for *α_β_*. Decreasing parameter beta indicates a “shear structural deformation”. This type of thermal deformation is typical for monoclinic crystals, as its thermal expansion is determined primarily by the change of the *β* angle. The angle between the 1st axis of the tensor and the crystallographic *a* axis decreases with the increase in the temperature; as a result, the structure expands mostly in the *α*_11_ direction, approaching the bisector of the *β* angle (Figure 8b). Thermal expansion tensors of brannerite at different temperatures are shown in Figure 8a. The decreasing *β* angle value is demonstrated by the rotation of the crystallographic *a* axis towards the *c* axis.

Note, that the expansion along the *a* axis is higher, compared to *α_c_*, at temperatures below 500 °C; afterwards, *α_c_* exceeds *α_a_* in the range 500–1100 °C. The maximum expansion occurs along or close to the direction of the longer (weaker) bonds. The U-O bond is more sensitive to the temperature increase. The expansion of the UO_6_ octahedra is more pronounced at the first stages of the heating. Its expansion drives the enlargement of the interlayer distance as UO_6_ octahedra link two TiO_6_ octahedral layers.

### 3.5. Thermal Analysis

The results of the Differential Scanning Calorimetric (DSC) experiment and Thermal Gravimetric Analysis (TGA) of the metamict brannerite are shown in Figure 9. The DSC curve indicates two thermal transitions in the studied temperature range. With the temperature increases, metamict brannerite starts to recrystallize, which results in an exothermic peak at 670 °C. The endothermic peak at 1082 °C shows that the sample undergoes partial melting. The X-ray diffraction pattern of the studied sample after the thermal analysis and complete cooling indicates the formation of rutile.

The TG curve shows a mass decrease of 0.11% from 99 to 307 °C, corresponding to the loss of adsorbed water from the sample surface. The partial oxidation of Fe and U results in the TG curve upward trend from 444 to 1055 °C. The mass of the sample increases by 0.27% and 0.45% in the temperature ranges 444–643 °C and 670–834 °C, respectively. A pause in the mass increase is observed in the range 643–670 °C, which implies several oxidation stages. The TG curve also demonstrates a slow mass increase by 0.17% in the range 834–1055 °C.

Between 1055 and 1115 °C, a mass loss of 0.60% is observed, which corresponds to the endothermic effect with a maximum at 1081 °C.

To conclude, there is an increase in a sample mass due to the oxidation between 444 and 1055 °C. A strong exothermic process associated with the crystallization of brannerite is observed at 670 °C. The mass increase almost stops at the same time and continues only after the completion of the crystallization of brannerite. Brannerite starts to partially decompose after 1000 °C.

### 3.6. Measurements of the Radioactivity

The activity of the radionuclides ^232^Th, ^226^Ra, ^137^Cs and ^40^K was measured.

The activity of the ^137^Cs and ^40^K isotopes was below the detection limit. The average value of ^226^Ra activity was 10,900 Bq/g, which accounts for up to 96% of the total radioactivity; the total gamma activity of the sample was 11,350 Bq/g.

The radioactivity was measured again after the annealing in air at 1100 °C. The specific activity of ^226^Ra decreased by 64% to 3396.07 Bq/g and the activity of ^232^Th is 716.58 Bq/g. The total is 4957.59 Bq/g.

Overall, these radioactive isotopes all belong to the uranium series. The heating procedure provokes the decrease of the radioactivity. It likely results from the escaping of some volatile isotopes from the sample, primarily radon. Radon may escape from the solid matrix at room temperature too; however, its emanation rates vary widely and are affected by the temperature, fission tracks and alpha-radiation damage. The small grain size of the powder sample promoted the radon diffusion.

## 4. Discussion and Concluding Remarks

Natural metamict brannerite from Akchatau, Kazakhstan, and its annealed sample were studied by EPMA and Raman spectroscopy to determine its chemical composition. The XRD and Raman spectra of the metamict brannerite indicated its structure to be heavily damaged. Iron and uranium are inhomogeneously distributed in the mineral due to the radiation damage. The studied metamict mineral grains show some zones enriched by the hydroxyl and SiO_2_. Pyrochlore and galena are found as inclusions. The crystal structure of brannerite is restored after annealing. The thermally processed brannerite’s texture is porous and homogeneous. 

The radioactivity of pristine and annealed samples of brannerite was measured. The specific activity of ^226^Ra accounts for up to 96% of the total radioactivity, which indicates that the main type of a decay in the mineral sample belongs to the uranium series. The heating process results in the loss of nearly half of the radionuclides in brannerite.

XRD at high temperatures was performed for the metamict sample to investigate the recrystallization process in situ. It has been reported previously in several works that brannerite recrystallizes in the temperature range of 800–1000 °C [7,19,21,25]. In contrast, the studied brannerite from Akchatau regains its structure at a temperature ~650 °C, approved also by the DSC measurements. We suggest that the recrystallization temperature is dependent mainly on the chemistry and homogeneity of the sample; but also depends on the degree of the radiation damage of the sample. Brannerite from Akchatau is characterized by the absence of the significant amounts of REE and yttrium typically observed at many other localities. The experimental conditions, such as a heating rate and annealing step, may also affect the recrystallization temperature. The HTXRD is recommended as a successful approach to identify, in detail, the recrystallization of the metamict minerals.

HTXRD was also performed on the annealed recrystallized brannerite. Thermal expansion of brannerite has been determined for the first time. The brannerite structure expands anisotropically with the temperature increase. The anisotropy is determined mainly by the change of the unit cell angles typical for monoclinic crystals. Despite its anisotropy, thermal expansion in brannerite is rather weak.

Considering that the waste immobilization form is usually a ceramic with various complex oxides, we can neglect the expansion anisotropy of single crystals when discussing the thermal expansion of the material. It is therefore reasonable to compare the volumetric CTE in brannerite with the available data for the other oxide materials. The *Ln*_2_Zr_2_O_7_ and *Ln*_2_Hf_2_O_7_ pyrochlores have volumetric expansion coefficients of 23.4–31.8 × 10^−6^ °C^−1^ in the range 25–1400 °C [26]. The CePO_4_ monazite expands very weakly, and its *α_v_* varies only slightly 21.0–27.4 × 10^−6^ °C^−1^ in the range 25–700 °C [27]. Perovskite CaTiO_3_ expands in a broader range: 37.87–55.37 × 10^−6^ °C^−1^ in the temperature range 25–1000 °C [28]. The volumetric coefficients of the thermal expansion of zirconolite-3*T* CaZrTi_2_O_7_ are 29.49–31.99 × 10^−6^ °C^−1^ in the same temperature range [28]. Brannerite has a low CTE at room temperature—*α_v_* = 16 × 10^−6^ °C^−1^, which increases up to 39.4 × 10^−6^ °C^−1^ at 1100 °C. In general, the thermal stability of brannerite is comparable to that of the other oxide radioactive waste-immobilizing matrices. Brannerite seems to be a promising material for such applications.

## Figures and Tables

**Figure 1 materials-16-01719-f001:**
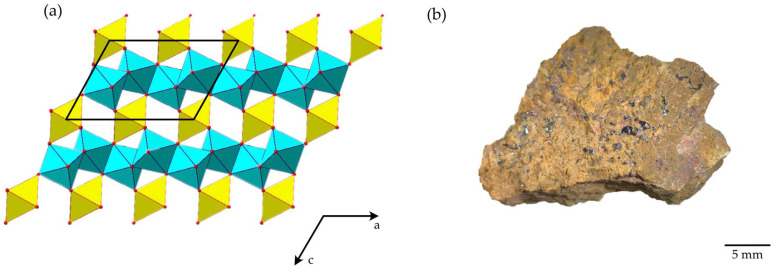
General projection of the crystal structure of brannerite along the *b* axis (UO_6_ = yellow octahedra; TiO_6_ = blue octahedra; structure drawing is based on data reported in Szymanski and Scott [2]) (**a**). Brannerite (black massive grains) in a host rock from Akchatau (Kazakhstan) (**b**).

**Figure 2 materials-16-01719-f002:**
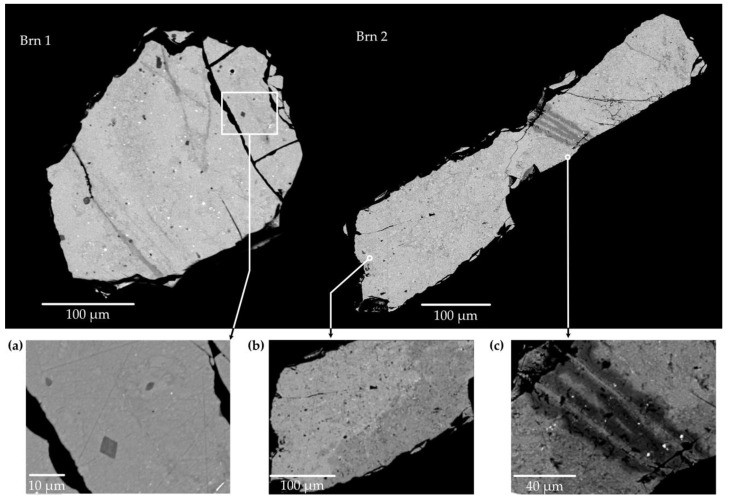
Two pristine grains of brannerite contain some impurities; enlarged areas of grains are shown in (**a**–**c**). The rhombic crystal of pyrochlore can be seen in (**c**). The altered region is well-represented in grain “***Brn2***”.

**Figure 3 materials-16-01719-f003:**
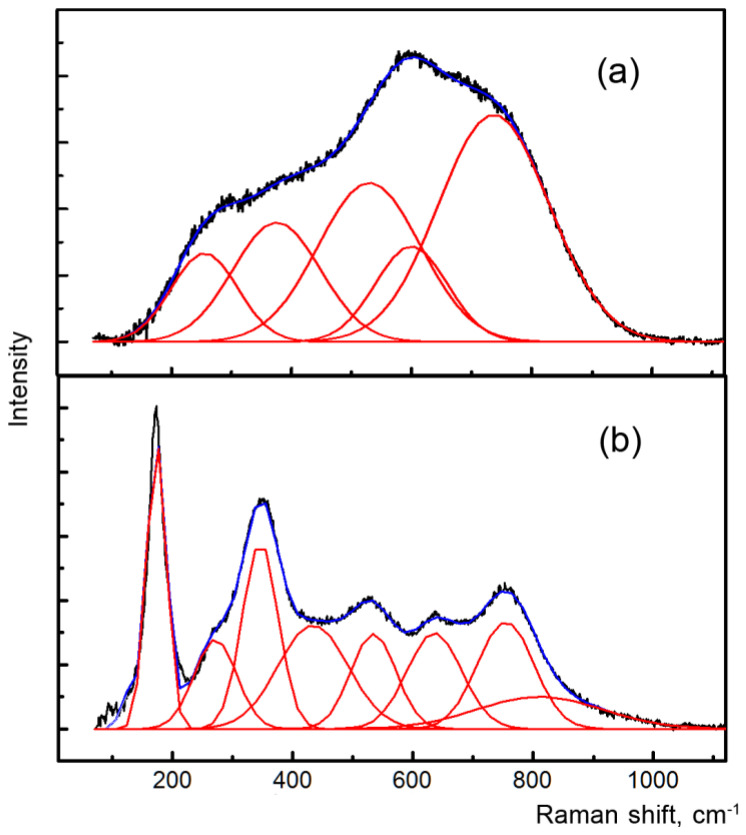
Raman spectra of the metamict (**a**) and annealed brannerite (**b**).

**Figure 4 materials-16-01719-f004:**
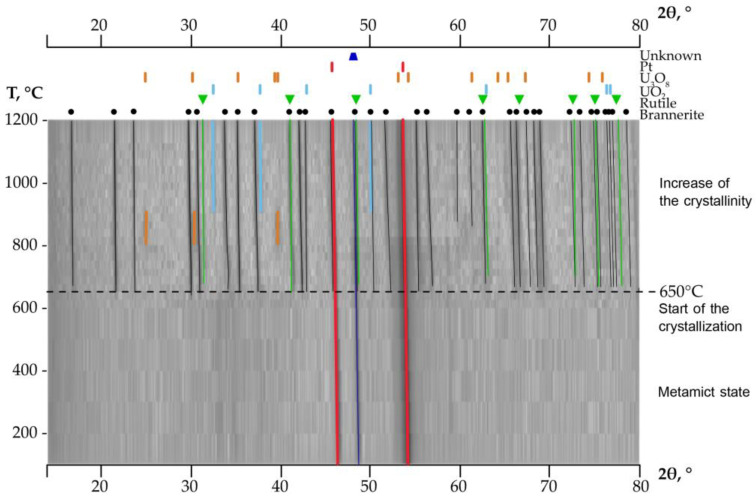
Evolution of X-ray powder diffraction patterns of the metamict brannerite in the range 100–1200 °C. There is no evidence of recrystallization until 625 °C.

**Figure 5 materials-16-01719-f005:**
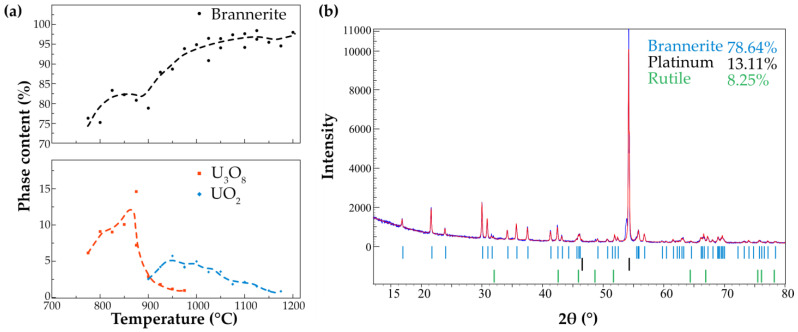
The content of the uranium oxide phases formed during the heating of metamict brannerite (**a**). XRD pattern obtained after cooling (**b**).

**Figure 6 materials-16-01719-f006:**
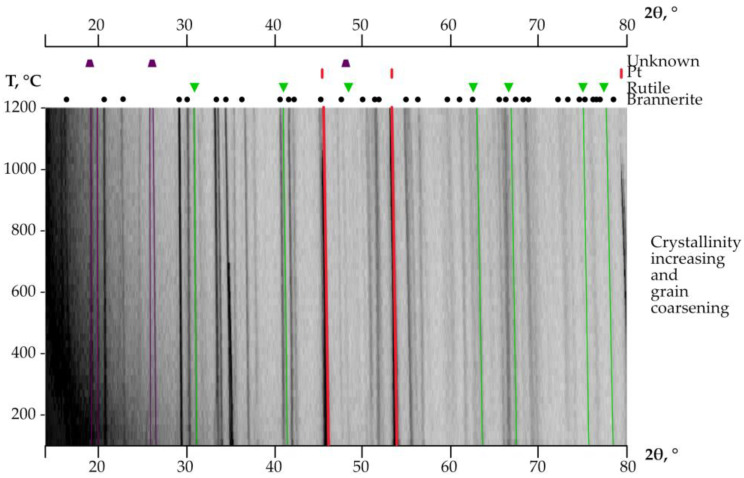
High-temperature XRD of the annealed brannerite. The admixture phases crystallized during the heating of the initial metamict brannerite are indicated above.

**Figure 8 materials-16-01719-f008:**
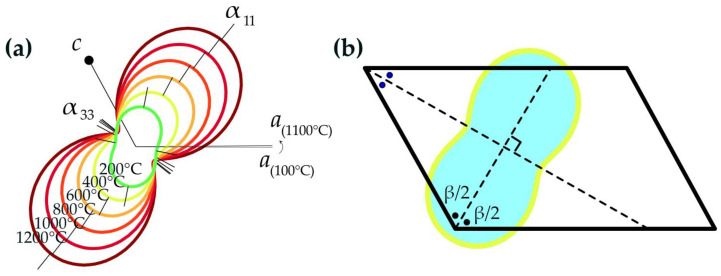
Pole figures of the thermal expansion coefficients of brannerite (**a**) and the projection on the unit cell (**b**). The *β* angle decrease is accompanied by the expansion along the *α*_11_ direction, which is a bisector of an obtuse angle.

**Figure 9 materials-16-01719-f009:**
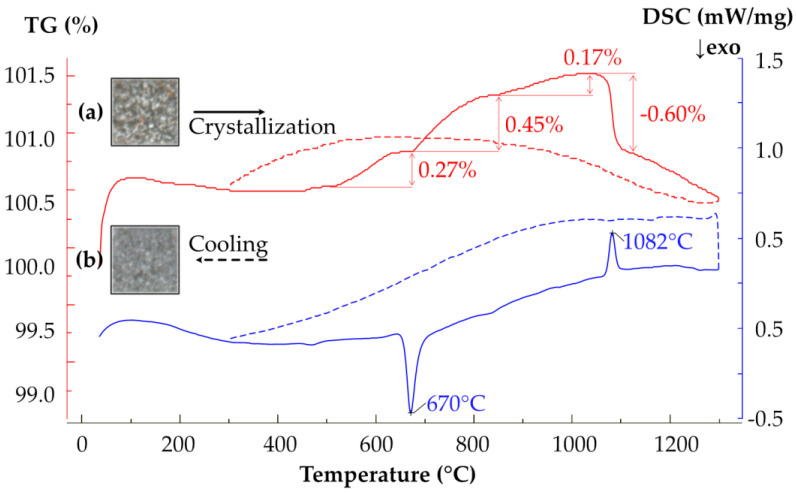
TG and DSC curves of metamict brannerite heated to 1300 °C. Photographs of the powder sample before (a) and after (b) thermal analysis are shown (field of view of each frame is 0.5 × 0.5 mm). Curves obtained during the cooling are shown by the dotted lines.

**Table 1 materials-16-01719-t001:** Composition (EPMA, wt.%) of the studied brannerite grains (***Brn1***, ***Brn2*** and ***Brn3***). Mineral formulas were calculated on the basis of “*B*” = 2 or “O” = 6 per formula unit.

	*Brn1* *-* *Unheated*			*Brn2-Unheated*			*Brn3-Heated*		
Oxides (Wt.%)	Average	Range	Deviation	Average	Range	Deviation	Average	Range	Deviation
TiO_2_	35.66	35.36–36.02	0.2	35.35	32.78–36.17	0.76	37.28	36.2–37.89	0.68
Fe_2_O_3_	0.68	0.25–0.75	0.21	0.86	0.25–1.37	0.53	0.26	0.11–0.36	0.09
Nb_2_O_5_							0.47	0.37–0.54	0.09
SiO_2_	0.3	0.3–0.3	/	1.45	0.77–1.73	0.39	0.14	0–0.22	0.07
UO_2_	56.22	55.35–57.55	0.97	56.11	51.88–58.29	2.05	57.07	51.44–59.38	2.92
ThO_2_	5.35	3.86–6.17	0.74	4.89	3.86–6.53	0.62	6.2	4.04–12.51	3.21
CaO	0.67	0.37–1.02	0.2	0.87	0.37–1.57	0.37	1.02	0.78–1.21	0.19
Na_2_O				0.52	0.46–0.56	0.05			
PbO	1.06	0.64–1.43	0.22	0.72	0.33–1.17	0.26	0.45	0.36–0.68	0.12
Total	99.56	99–100.26		98.89	97.08–100.11		102.65	101.28–104.47	
Formula coefficient	“*B*” = 2	“O” = 6		“*B*” = 2	“O” = 6		“*B*” = 2	“O” = 6	
Ti	1.94	1.92		1.86	1.86		1.97	1.95	
Fe^3+^ *	0.04	0.02		0.04	0.02		0.01	0.01	
Nb							0.01	0	
Si	0.02	0.02		0.1	0.1		0.01	0.01	
∑=	2	1.96		2	1.98		2	1.97	
U	0.91	0.9		0.87	0.87		0.89	0.88	
Th	0.09	0.09		0.08	0.08		0.09	0.09	
Ca	0.05	0.05		0.06	0.06		0.07	0.07	
Na				0.07	0.03				
Pb	0.02	0.02		0.01	0.01		0.01	0.01	
∑=	1.06	1.05		1.09	1.06		1.07	1.06	
O	6.06	6		5.99	6		5.2	6	
□+	/	0.12		/	0.28		/	0.05	

* All iron is calculated as trivalent.

**Table 2 materials-16-01719-t002:** The chemical composition of inclusions in ***Brn1*** and ***Brn2***.

	S	CaO	TiO_2_	PbO	ThO_2_	UO_2_	Normalized Total
** *Brn1* **	17.05	0.25	9.66	50.29	2.69	20.06	100
** *Brn2* **	20.88	0.21	5.88	60.35	0.91	12.15	100

**Table 3 materials-16-01719-t003:** Raman bands and their assignment in the annealed studied brannerite sample and the reference data from [18,19].

This Study	Reference Data			
Annealed *Brn3* Sample	Annealed Brannerite [17]		Natural Annealed Brannerite [18]	
Raman Shift (cm^−1^)	Raman Shift (cm^−1^)	Assignment	Raman Shift (cm^−1^)	Assignment
173	171	O–Ti–O bending modes	178	Ti–O bending modes
271	269	U_3_(OH)_3_ out-of-plane bending vibrations	263	O–Ti–O symmetric stretching vibrations
346	356	U_3_(OH)_3_ or U_2_O(OH) group elongation vibrations	339	O–Ti–O vibrations
434 534	400–600	Ti–O chain vibrations or U_3_O bridge elongation	400–600	U–O vibrations
636	641	Ti–O symmetric stretching vibration	641	Ti–O symmetric stretching vibration
754 811	766 811	(UO_2_)^2+^ symmetric stretching vibrations	743 824	O–Ti–O symmetric stretching vibrations (UO_2_)^2+^ symmetric stretching vibrations

**Table 4 materials-16-01719-t004:** Principal values of the thermal expansion tensor and volumetric expansion of brannerite.

T, °C	*α ×* 10^6^ °C^−1^				
	100	300	600	900	1100
*α* _11_	8.39 (67)	12.03 (42)	18.19 (33)	24.44 (93)	28.6 (1.4)
*α*_22_ = *α_b_*	4.59 (37)	4.88 (17)	5.304 (95)	5.73 (22)	6.01 (30)
*α* _33_	4.38 (35)	4.94 (17)	5.013 (89)	4.91 (19)	4.79 (24)
*∠*(*α*_11_, *α_a_*)	78.9°	61.8°	54.4°	51.7°	50.6°
*∠*(*α*_33_, *α_c_*)	50.2°	33.2°	26.0°	23.5°	22.7°
*α_a_*	4.53 (56)	6.52 (31)	9.48 (21)	12.42 (57)	14.36 (83)
*α_c_*	6.75 (74)	7.07 (41)	7.54 (28)	8.01 (74)	8.3 (1)
*α_β_*	−1.32 (70)	−3.00 (40)	−5.53 (23)	−8.07 (63)	−9.78 (94)
*α_V_*	17.4 (1.3)	21.85 (75)	28.51 (51)	35.1 (1.3)	39.4 (1.9)

## Data Availability

The data can be provided by the authors upon request.

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
