# Peer review of "The Chemistry, Recrystallization and Thermal Expansion of Brannerite from Akchatau, Kazakhstan"

_materials, 2023, doi:10.3390/ma16041719_

Round 1

Reviewer 1 Report

Regarding the Work “The chemistry, recrystallization and thermal expansion of brannerite from Akchatau, Kazakhstan”. The hypothesis in general terms is adequate, but it is necessary to consider the following:

• It is necessary to adjust to the format of the figures and tables in general.

• The X-ray spectrum of figure 5 does not have the PDFs, it is worth mentioning that only two lines can be seen for platinum and rutile, it is not seen clearly either; Thus, it is necessary to expand the information of the Ritvelt analysis. On the other hand, it would be convenient to present the crystal size and crystallinity data, in addition to marking the expansion or contraction of the peaks in the spectra.

• In figure 4, it is recommended to put all the diffractograms of the different temperatures to be able to visualize the present alterations.

• In figure 6, it is recommended put all the diffractograms of the different temperatures to be able to visualize the phases present.

• It is recommended that in figure 2 the impurities present be indicated in the microphotographs.

• It is recommended to do the discussion and conclusions section separately.

·         It is recommended to update the bibliographical citations.

Author Response

Rev. # 1

Regarding the Work “The chemistry, recrystallization and thermal expansion of brannerite from Akchatau, Kazakhstan”. The hypothesis in general terms is adequate, but it is necessary to consider the following:

  • It is necessary to adjust to the format of the figures and tables in general.

Reply: Thank you! The format was adjusted

  • The X-ray spectrum of figure 5 does not have the PDFs, it is worth mentioning that only two lines can be seen for platinum and rutile, it is not seen clearly either; Thus, it is necessary to expand the information of the Ritvelt analysis. On the other hand, it would be convenient to present the crystal size and crystallinity data, in addition to marking the expansion or contraction of the peaks in the spectra.

Reply: We have modified Figure 5b. Now the line are clearly visible. The peaks are quite narrow in the represented X-ray pattern which is the sign of the well crystallized mineral.

  • In figure 4, it is recommended to put all the diffractograms of the different temperatures to be able to visualize the present alterations.

Reply: Yes, indeed, all of the diffractograms at different temperatures are merged in Figure 4.

  • In figure 6, it is recommended put all the diffractograms of the different temperatures to be able to visualize the phases present.

Reply: Yes, indeed, all of the diffractograms at different temperatures are merged in Figure 4.

  • It is recommended that in figure 2 the impurities present be indicated in the microphotographs.

Reply: Yes, the impurities, including crystals, are clearly visible in Figure 2.

  • It is recommended to do the discussion and conclusions section separately.

Reply: We have modified part 4. Discussion and concluding remarks

       It is recommended to update the bibliographical citations.

Reply: Thank you very much! The reference list was updated.

Reviewer 2 Report

Dear Editors, 

the manuscript deals with brannerite, a very interesting, in many aspects,  radioactive mineral.  This work is a mineralogical, chemical and crystallographic analysis on metamict and annealed grains using a wide range of analytical instruments.

The manuscript is well written in a comprehensive way with good structure. 

Some comments: 

1. EMPA analyses: Usually natural brannerite contains Lanthanides, Y and Al which are not measured and probably may explain the low Total conc. in some samples. Is this the reason for the darker and brighter parts of the crystals?

2. Also the Total average for the annealed brannerite sample is quite high (over 102%).  You could you add an explanation for this?

3. It is not clearly explained why the recrystallization temperature from your samples is lower than these from other localities. 

4. The last chapter is not very conclusive.

Kind regards

Author Response

Rev. # 2

Dear Editors, 

the manuscript deals with brannerite, a very interesting, in many aspects,  radioactive mineral.  This work is a mineralogical, chemical and crystallographic analysis on metamict and annealed grains using a wide range of analytical instruments.

The manuscript is well written in a comprehensive way with good structure. 

Some comments: 

  1. EMPA analyses: Usually natural brannerite contains Lanthanides, Y and Al which are not measured and probably may explain the low Total conc. in some samples. Is this the reason for the darker and brighter parts of the crystals?

Reply: Yes, studied sample of brannerite does not contain lanthanides and Al in detectable by microprobe amounts. We attach the obtained spectra.

  1. Also the Total average for the annealed brannerite sample is quite high (over 102%).  You could you add an explanation for this?

Reply: There are probably differences between the calculated valence of the element and the actual valency (U, Fe) after the recrystallization, resulting in a slight deviation in wt, % content.

  1. It is not clearly explained why the recrystallization temperature from your samples is lower than these from other localities. 

Reply: The following text was added to the manuscript: We suggest that the recrystallization temperature is dependent mainly on the chemistry and homogeneity of the sample; but also depends on the degree of the radiation damage of the sample. Brannerite from Akchatau is characterized by the absence of the significant amounts of REE and yttrium, typically observed at many other localities. The experimental conditions, such as a heating rate, and annealing step, may also affect the recrystallization temperature.

  1. The last chapter is not very conclusive.

 Reply: We have modified the conclusion part.

Reviewer 3 Report

Dear Editor-in-Chief

Reverently, in the paper entitled “The chemistry, recrystallization and thermal expansion of bran-nerite from Akchatau, Kazakhstan”, the authors have attempted to explain the object in detail, so they brought more information in this regard. However, some improvements and clarifications in the manuscript as below are necessary to be applied. 

-          The conceptual relationship between the sentences in the introduction is weak and needs to be revised.

-          The introduced symbols (e.g. a, b, …) should be explained or referred at first.

-          In the introduction, the necessity and objectives of the work must be specified and presented.

-          Provided explanation of the unheated sample composition (page 5, line 163) is better to depict in a table.

-          The author has not given any explanation about the specs of the setup used (heating system, …) for the experiments

-          In Fig. 3, all significant and meaningful variations should be specified.

-          The author should explain more about the reasons behind different crystallization temperature points. In references cited (8, 20, 21) some reasons have been proposed.

-          the range of calculated R-factors (page 9, 6.2-12.6%) is a bit wide. The reasons should be mentioned.

Author Response

Rev #3

Reverently, in the paper entitled “The chemistry, recrystallization and thermal expansion of bran-

nerite from Akchatau, Kazakhstan”, the authors have attempted to explain the object in detail, so

they brought more information in this regard. However, some improvements and clarifications in the manuscript as below are necessary to be applied.

- The conceptual relationship between the sentences in the introduction is weak and needs to be revised.

Reply: Thank you! We have modified the Introduction

- The introduced symbols (e.g. a, b, ...) should be explained or referred at first.

Reply: Symbols for the unit-cell parameters are generally accepted and do not require further explanation.

- In the introduction, the necessity and objectives of the work must be specified and presented.

Reply: The aims were added.

- Provided explanation of the unheated sample composition (page 5, line 163) is better to depict

in a table.

Reply: The line was modified

- The author has not given any explanation about the specs of the setup used (heating

system, ...) for the experiments

Reply: Added.

- In Fig. 3, all significant and meaningful variations should be specified.

Reply: The figure was redrawn.

- The author should explain more about the reasons behind different crystallization temperature

points. In references cited (8, 20, 21) some reasons have been proposed.

Reply: The following text was added to the manuscript: We suggest that the recrystallization temperature is dependent mainly on the chemistry and homogeneity of the sample; but also depends on the degree of the radiation damage of the sample. Brannerite from Akchatau is characterized by the absence of the significant amounts of REE and yttrium, typically observed at many other localities. The experimental conditions, such as a heating rate, and annealing step, may also affect the recrystallization temperature.

- the range of calculated R-factors (page 9, 6.2-12.6%) is a bit wide. The reasons should be

mentioned.

Reply: The R-factors calculated by the Rietveld refinement increase with temperature, and fluctuate in the range of 6.2-12.6%. High temperatures cause atoms' thermal vibration to rise, which reduces the refinement quality.  

Round 2

Reviewer 3 Report

The current form of the manuscript is acceptable for publication.